# Noncoding AUG circRNAs constitute an abundant and conserved subclass of circles

Lotte VW Stagsted, Katrine M Nielsen, Iben Daugaard, Thomas B Hansen

**Circular RNAs (circRNAs) are a subset of noncoding RNAs previously considered as products of missplicing. Now, circRNAs are considered functional molecules, although to date, only few functions have been experimentally validated. Here, based on RNA sequencing from the ENCODE consortium, we identify and characterize a subset of circRNAs, coined AUG circRNAs, encompassing the annotated translational start codon from the protein-coding host genes. AUG circRNAs are more abundantly expressed and conserved than other groups of circRNAs, and they display flanking sequences that suggest an *Alu*-independent mechanism of biogenesis. The AUG circRNAs contain part of bona fide open reading frame, and in the recent years, several studies have reported cases of circRNA translation. However, using thorough cross-species analysis, extensive ribosome profiling, proteomics analyses, and experimental data on a selected panel of AUG circRNAs, we observe no indications of translation of AUG circRNAs or any other circRNAs. Our data provide a comprehensive classification of circRNAs and, collectively, the data suggest that the AUG circRNAs constitute an abundant subclass of circRNAs produced independently of primate-specific *Alu* elements.**

## Introduction

Noncoding RNAs (ncRNAs) constitute the vast majority of the human transcriptome as only a few percent of the produced transcripts are translated into proteins (ENCODE Project Consortium, 2012). NcRNAs represent a highly heterogeneous group of molecules that besides including essential elements of protein synthesis, ribosomal RNA and tRNA, also comprise small RNAs, such as microRNAs (miRNAs), which are involved in regulation of mRNA stability and protein synthesis (Bartel, 2009), as well as long noncoding RNAs (lncRNAs) with many established functionalities (Böhmdorfer & Wierzbicki, 2015; Noh et al, 2018). Recently, by means of high throughput tools, circular RNAs (circRNAs) were added to the rapidly expanding list of ncRNA (Ebbesen et al, 2016). CircRNAs

are typically derived from annotated protein-coding genes, but because of their relatively low abundance compared with their linear mRNA counterparts, circRNA molecules were first presumed to be missplicing events of the spliceosome with little to no relevance (Cocquerelle et al, 1993; Zaphiropoulos, 1997). Although this may be the case for a substantial subset of circRNAs, the identification and functional characterization of the highly conserved circRNA and miR-7-sponge, CDR1as/ciRS-7 (Hansen et al, 2013b; Memczak et al, 2013), and extensive profiling of differentially expressed circRNAs from RNA sequencing analyses (Salzman et al, 2012; Memczak et al, 2013; Rybak-Wolf et al, 2015; Veno et al, 2015) strongly support circRNAs as biologically relevant RNA species in eukaryotic cells. CircRNAs are generated by nonlinear splicing (coined backsplicing) where an upstream splice acceptor (SA) is covalently joined to a downstream splice donor (SD) resulting in a circular structure (Hansen et al, 2011; Jeck et al, 2013). This results in a very high intracellular stability due to the lack of free ends, which protects them from normal exonucleolytic decay. CircRNAs are mostly composed of exonic regions (most commonly 2–3 exons) derived from annotated protein-coding transcripts (Zhang et al, 2014). The current model of biogenesis suggests that backsplicing is stimulated by bringing the involved splice sites into close proximity (Ebbesen et al, 2016). This is conventionally facilitated by inverted *Alu* elements (IAEs) (Jeck et al, 2013; Zhang et al, 2014); however, trans-acting RNA-binding factors have also been implicated in circRNA formation (Ashwal-Fluss et al, 2014; Conn et al, 2015; Li et al, 2017).

With the exception of the exon–intron circRNAs (Li et al, 2015), circRNAs are exported to the cytoplasm (Jeck et al, 2013) with a recent study reporting the *Drosophila* Hel25E and its human homologs to regulate nuclear export of circRNAs in a length-dependent manner (Huang et al, 2018). In the cytoplasm, circRNAs have been shown to tether and "sponge" miRNAs, initially exemplified by CDR1as/ciRS-7 harbouring >70 miR-7–binding sites (Hansen et al, 2013a; Memczak et al, 2013). Since then, several other examples have been published showing anti-miR effects of circRNA expression (Peng et al, 2016; Zheng et al, 2016; Chaiteerakij et al, 2017), although bioinformatics analysis indicates that—apart from ciRS-7—miRNA-binding sites are generally not enriched in circRNA more than expected by chance (Guo et al, 2014). CircRNAs can also sequester RNA binding proteins

Department of Molecular Biology and Genetics, and Interdisciplinary Nanoscience Center (iNANO), Aarhus University, Aarhus, Denmark

Correspondence: tbh@mbg.au.dk

and hereby modulate protein activity (Ashwal-Fluss et al, 2014). In addition, synthetic circRNAs have been engineered to express protein by the use of internal ribosome entry sites allowing cap-independent translation (Wang & Wang, 2015). Recently, it was shown that open reading frames (ORFs) within endogenously expressed circRNAs give rise to circRNA-specific peptides (Legnini et al, 2017; Pamudurti et al, 2017; Yang et al, 2017, 2018; Zhang et al, 2018a, 2018b), suggesting that circRNAs are not necessarily exclusively noncoding.

In this study, publicly available RNA sequencing datasets from the ENCODE consortium are used to characterize the circRNA transcriptomes in 378 human and 75 murine samples, and the most abundant circRNAs in each dataset are identified, analyzed, and stratified based on their genomic features. These analyses reveal that a substantial fraction of highly abundant circRNAs derives from exons encoding the translational start codon, here coined AUG circRNAs. In addition, the AUG circRNAs are more conserved than other groups of circRNAs and generally rely on an IAE-independent mode of biogenesis. Last, to determine the protein-codon ability of AUG circRNAs, we conduct extensive analyses of cross-species conservation and ribosome profiling (RiboSeq). This shows that ORF-associating features are not preserved in evolution and that backsplice-spanning reads found in RiboSeq datasets are not derived from translating ribosomes. Consistently, we fail to confidently detect any peptides derived from AUG circRNAs by mass spectrometry or by ectopic overexpression in cell lines. Collectively, these results suggest that circRNAs are generally not subjected to translation and, thus, the functional relevance of the most conserved and abundant AUG circRNAs remains elusive.

## Results

### The ENCODE circRNA landscape

To obtain a comprehensive overview of circRNA expression across multiple tissues and cell lines, we took advantage of the total RNA sequencing datasets on human and mouse samples made available from the ENCODE consortium (see Table S1). We conducted circRNA prediction and quantification using two established pipelines; find_circ (Memczak et al, 2013) and circexplorer2 (Zhang et al, 2016a). In total, find_circ and circexplorer2 identify 140,304 and 235,179 unique circRNAs using slightly modified settings (see methods), respectively, of which 81,589 are shared by both algorithms (Fig 1A). The notable fraction of circRNAs only predicted by one algorithm—the so-called exotic circRNAs—is in general lowly expressed (Fig 1B), which is also reflected by a small subset of exotic circRNAs in the top 1,000 expressed circRNA candidates predicted by each algorithm (1–8%, data not shown). Consistently, we observe a high positive correlation between the algorithms for the abundant circRNA species (Fig 1C, similar analyses for mouse samples are shown in Fig S1A–C). We have previously shown that exotic circRNAs are more likely to be false positives (Hansen et al, 2015), and, therefore, we decided to focus only on the circRNAs jointly predicted by both algorithms.

The used ENCODE data comprise 378 samples derived from 218 different human tissues and cell lines (or 75 samples from 26

tissues in mouse, Tables S1 and S2). Plotting the expression of circRNAs in each sample reveal a marked difference in circRNA expression between the samples with particular abundant circRNA expression levels in tissues (Figs S2 and S4A), whereas the similar analysis for mRNAs show similar expression in all biosample types (Figs S3 and S4B). Moreover, even though the detected diversity of circRNAs is much lower in mouse, the overall expression levels are comparable (Fig S5A). CircRNA levels have previously been correlated with proliferation, that is, circRNAs tend to accumulate in slow or non-proliferative tissue (Bachmayr-Heyda et al, 2015). Thus, circRNA profiling from nondividing cells may dominate the average expression levels of circRNA across samples. Instead of comparing expression across samples, we instead focused on the highest expressed circRNA in each sample (the α-circRNA). Here, the α-circRNA in many samples exhibits disproportionally high expression compared with the bulk of circRNAs. In fact, assuming a log-normal distribution of circRNA expression, the α-circRNAs are significant outliers in more than half of the samples (246 of 378, fdr < 0.05, one-tailed Grubbs test), whereas only 4 of 378 samples show similar significant outlier mRNAs. This tendency is also observed in mouse (Fig S5B).

Based on the ENCODE data, circHIPK3 is the most predominant α-circRNA followed by the miR-7 sponge, ciRS-7 (Fig 1D). Even though most of the top 10 α-circRNAs are found in the mouse dataset, only circSLC8A1 and circCDYL are shared in the top 10 between mouse and human (Figs 1D and S6A).

We then zoomed in on the top 10 α-circRNAs, that is, the 10 circRNAs most often seen as the highest expressed in a given sample, to determine the genomic features associating with these highly abundant circRNA species (see Fig 1E). Here, the human α-circRNAs are flanked with very distal IAEs, which is in stark contrast to the bulk of circRNAs (Fig 1F) and the prevalent model of biogenesis (Jeck et al, 2013; Zhang et al, 2014). Moreover, no significant association between circRNA producing loci and other inverted repeat elements are observed for the α-circRNAs specifically or circRNAs in general compared with host gene exons (Fig 1F). In mouse, no repetitive elements are selectively demarcating circRNAs from host exons (Fig S6B), although a slight tendency towards proximal B1/alu SINE elements was detected. Instead, for both species, we observe a clear tendency for α-circRNAs to have very long flanking introns (Figs 1F and S6B, ciRS-7, circRMST, and circFAT3 are without annotated flanking introns and, thus, excluded in this analysis). Moreover, a positive correlation between intron length and IAE distance is detected (Figs 1G and S6C), indicating that circRNAs either use an *Alu*-dependent mechanism of biogenesis or require long flanking introns to favour backsplicing.

### AUG circRNAs are highly expressed and conserved

The vast majority of circRNAs derive from annotated splice sites (Zhang et al, 2016a), and we decided to stratify circRNAs by host gene annotation (see Fig 2A). Here, circRNAs derived from exons containing annotated start codons, coined AUG circRNA, comprise 5 of the top 10 α-circRNAs in both human and mouse samples, whereas the percentage of AUG circRNA in general is 7–11% (Figs 2B and S7A). In fact, the AUG circRNAs in both human and mouse also show a significant over-representation in subsets of highly

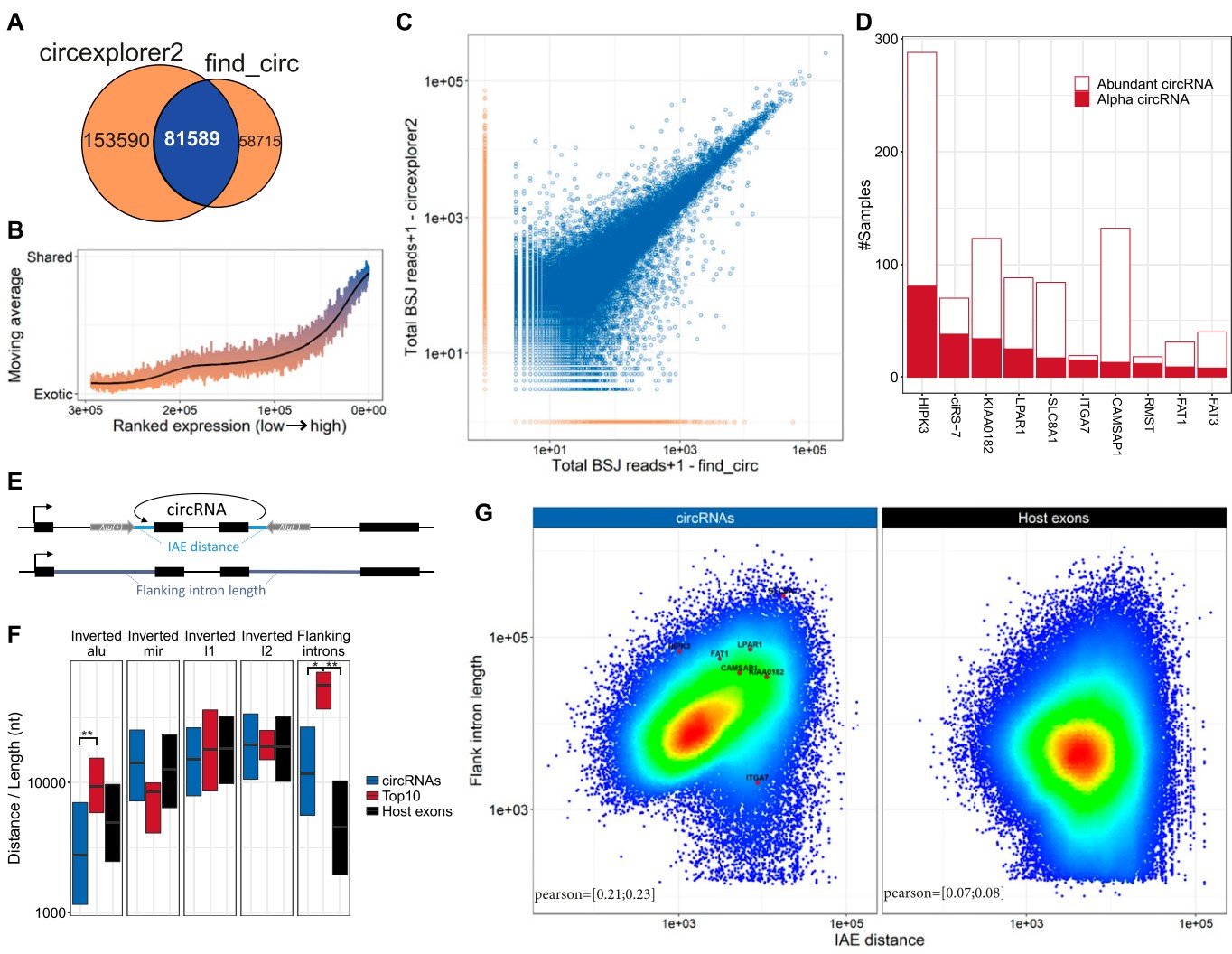

**Figure 1. Abundant circRNAs.**
**(A)** Venn diagram showing the number of exotic (orange) and shared (blue) circRNAs found by find_circ and circexplorer2 algorithms in the ENCODE datasets. **(B)** Smoothed fraction of shared circRNAs found by find_circ and circexplorer2 as a function of ranked expression. **(C)** Scatterplot depicting the number of BSJ-spanning reads obtained from find_circ and circexplorer2 across all the samples analyzed. The points are color-coded as shared (blue) or exotic (orange). **(D)** The $\alpha$ frequency of the top 10 most commonly found $\alpha$-circRNAs and the frequency of being an abundant circRNA (i.e., one of the top 10 expressed circRNAs in a sample) are plotted as a stacked bar plot. **(E)** Schematic illustration of the flanking intron length and IAE. **(F)** Boxplot comparing the distance to inverted repeat element and flanking intron length for circRNAs in general (n = 81,589), host gene exons (n = 131,002), and the top 10 $\alpha$-circRNAs. *$P < 0.05$; **$P < 0.01$, Wilcoxon rank-sum test. **(G)** Density-colored scatterplot showing relationship between IAE and flanking intron length for all circRNAs (left) and host gene exons (right). The top 10 $\alpha$-circRNAs are highlighted to the left.

expressed circRNAs, which is not seen for other circRNA subclasses (Figs 2B and S7A). Consistently, in both human and mouse, AUG circRNAs are generally and significantly more abundant than other circRNAs (Fig S8A–B and D–E) both in terms of absolute expression but also regarding the circular-to-linear ratios, whereas the host genes, from which the AUG circRNAs are derived, are not significantly more abundant than for other circRNAs (Fig S8C and F).

Using liftover (UCSC), we evaluated the number of human circRNAs re-identified in the mouse dataset of circRNAs as a measure of conserved biogenesis. Similar to AUG circRNAs, the fraction of conserved circRNAs increase with expression (Fig 2C). Focusing specifically on the top 1,000 most abundant human circRNAs based on total backsplice junction (BSJ)-spanning reads, 39% of all AUG circRNAs are conserved comprising almost twice as

many conserved species compared with the other circRNA subgroups (Fig 2D and $P = 9.4 \times 10^{-4}$, Fisher's exact test), despite the fact that the 5′UTR-embedded SAs in human AUG circRNAs are often (21%) not annotated as splice sites in mouse. As with the top 10 $\alpha$-circRNAs, AUG circRNAs generally exhibit distal IAE and longer flanking introns (Fig 2E and F). In fact, AUG circRNAs, conserved circRNAs and host gene exons exhibit an overall similar distribution of IAEs (Fig 2E). In contrast, the flanking intron lengths effectively demarcate AUG circRNA and conserved circRNAs from host gene exons (Fig 2F), which is also supported by the analysis of AUG circRNAs in mouse (Fig S7B). In fact, using an atlas of circRNAs from *Drosophila* (Westholm et al, 2014), AUG circRNAs also increase in frequency with circRNAs expression and associate with significantly longer introns compared with other circRNAs (Fig S7C and D),

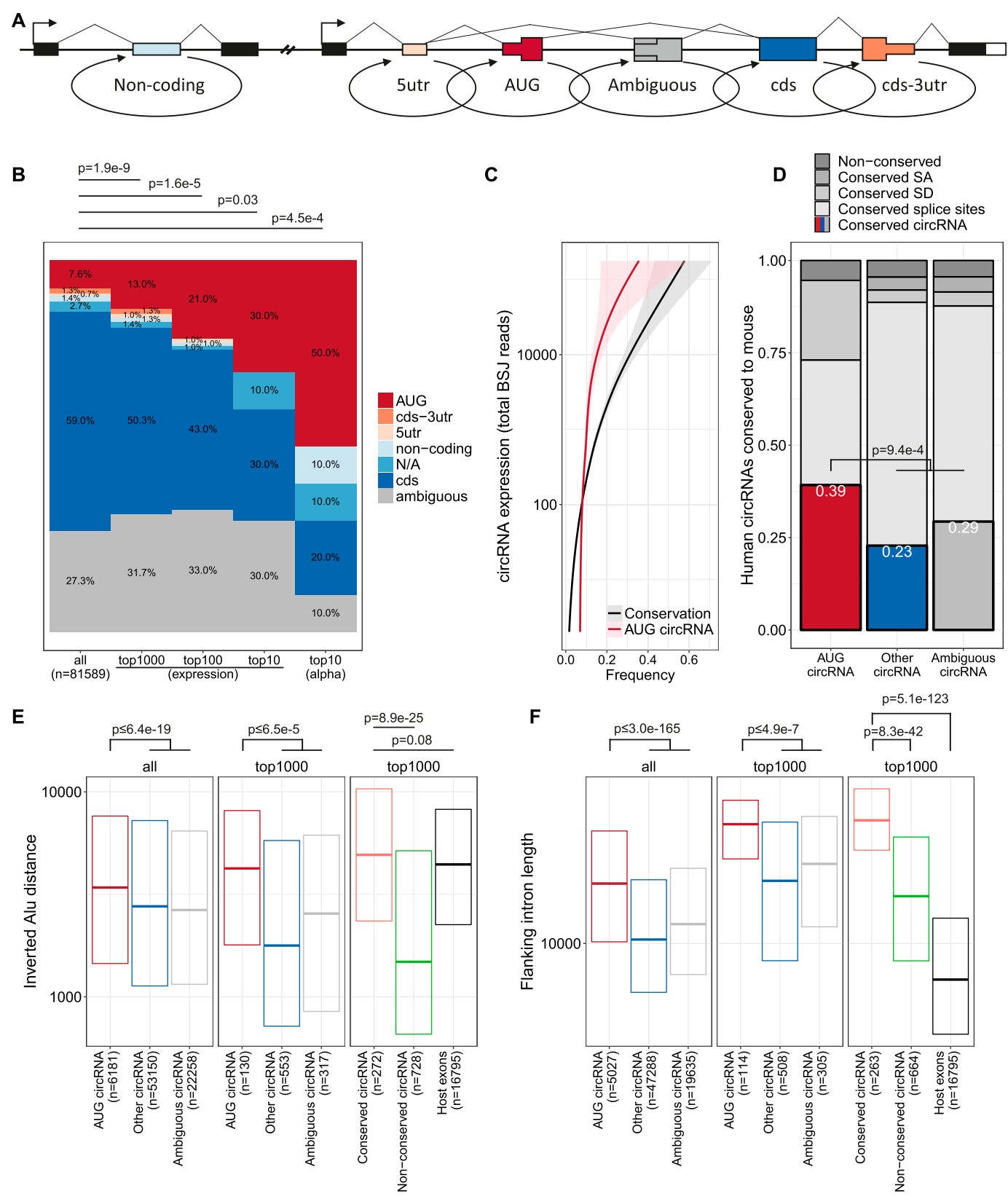

**Figure 2. The AUG circRNAs.**
**(A)** Schematics showing circRNA annotation. The genic features are not drawn to scale. **(B)** Frequency of circRNA-annotations for either all circRNA, the top 1,000, top 100, top 10 circRNAs based on overall expression (RPM), and the top 10 α-circRNAs, color-coded as denoted. *P*-values are calculated using Fisher's exact test. **(C)** Smoothed

whereas fruit fly circles in general show no evidence of flanking inverted elements (Westholm et al, 2014). Thus, based on the observations that AUG circRNAs are more conserved and overall devoid of flanking IAEs and the fact that AUG circRNA features may extent all the way to invertebrates, we propose that AUG circRNAs are more likely to be biologically relevant and to use an *Alu*-independent biogenesis pathway.

To demarcate *Alu*-dependent from *Alu*-independent circRNAs, we empirically determined the distance to nearest IAE by which the cumulative fraction of circularizing exons differed the most from noncircularizing host exons. Here, approximately 45% of all human circRNAs has an IAE within 2,300 nucleotides total distance, whereas this cutoff only applies to 22% of host exons (Fig S9A). We, thus, defined this 45% subset as the *Alu*-dependent circRNAs. Based on this demarcation, only 13% of *Alu*-dependent circRNAs are observed in mouse, which is consistent with the fact that *Alu* elements are primate specific (Fig S9C). In contrast, 41% of *Alu*-independent and 31% of circRNAs with long flanking introns (defined empirically as flanking intron >6,500 nts, Fig S9B) are conserved (Fig S9C), which suggests that at least for the evolutionary relevant circRNAs, biogenesis relies more on having long flanking introns instead of proximal inverted *Alu* repeats. However, the requirement for long flanking introns in circRNA biogenesis is currently unclear, and, therefore, the mechanism governing production of most abundant and conserved circRNAs remains undisclosed.

Recently, the RNA resolvase, DHX9, was shown to inhibit circRNA production by unwinding and destabilizing RNA structures formed by IAEs in flanking regions of circRNAs (Aktaş et al, 2017). DHX9 is proposed to protect cells from adverse secondary structures in the nucleus. As a consequence, circRNAs sensitive to DHX9 depletion are considered products of aberrant backsplicing mediated by random insertion of inverted repeat elements, and, therefore, these circRNAs are more likely to be functionally irrelevant. Based on RNAseq from DHX9-depleted HEK293 cells (Table S3), it is possible to determine the subset of circRNAs sensitive to DHX9 expression. As above, we identified circRNA expression using find_circ and circexplorer2 (Fig S10A–C), and we selected the top 1,000 expressed circRNAs from this analysis (Table S4). Here, roughly 25% (275 circRNAs of 1,000) responds significantly (fdr < 0.05) to the DHX9 depletion (Fig 3A), with a clear tendency towards proximal IAE and short flanking introns (Fig 3B and C). Consistently, 39% of circRNAs designated as *Alu* dependent (IAE distance <2,300 nt) are DHX9-sensitive compared with only 12% of the non-*Alu* circRNAs, whereas long flanking introns are generally insensitive to DHX9 compared with short introns (43 versus 23%, Fig S10D). Interestingly, in alignment with the analyses described above, AUG circRNAs are significantly reduced in the DHX9-sensitive fraction (5 versus 15%, $P = 4.8 \times 10^{7}$, Fig 3D). In fact, only 10% of the AUG circRNAs compared with 26% of non-AUG circRNAs is affected significantly in expression upon DHX9 depletion (Fig 3E).

CircHIPK3 has previously been characterized as an *Alu*-dependent circRNA (Zheng et al, 2016); however, it is also an AUG circRNA, as well as the overall highest expressed circRNA in the ENCODE data. As such, the most proximal IAEs are within 2,300 nt (see Fig 3B), and while these *Alu* elements could stimulate biogenesis, circHIPK3 carries more than 60 *Alu* elements in the immediate flanking introns and is insensitive to DHX9 depletion (Fig 3A). DHX9-insensitive biogenesis is also observed for the two additional top 10 expressed AUG circRNAs, circSETD3, and circVRK1. Instead, the three AUG circRNAs in the top 10 fraction all associate with very long flanking introns (Fig 3B), and notably, circZBTB44, which in this analysis is termed "ambiguous" because it overlaps both an AUG and a noncoding transcript, share the same features. Consistently, HITS-CLIP analysis of DHX9 occupancy (Table S3) shows a clear selection for binding in the immediate flanking regions of *Alu*-dependent compared with *Alu*-independent circRNAs (Figs 3F and S10E) but also a clear preference for non-AUG over AUG circRNAs (Figs 3F and S10F). Collectively, this strongly indicates that the AUG circRNAs are generally not affected by DHX9 helicase activity and, thus, not depending on IAE for biogenesis.

## No detectable protein production from ectopically expressed AUG circRNAs

Recent studies have shown that circRNAs despite lacking a 5′cap and 3′poly(A)-tail are still capable of recruiting ribosomes and act as templates for protein synthesis (Legnini et al, 2017; Yang et al, 2017; Pamudurti et al, 2017). This is most likely facilitated by internal ribosome entry site-like elements in the circRNA required for cap-independent translation. The AUG circRNAs all contain a 5′ part of a bona fide ORF, and translation of this putative ORF will in most cases produce a truncated protein mimicking the N-terminal part of the host gene encoded protein. To test the hypothesis that AUG circRNAs are in fact protein-coding circRNAs, we initially focused on the two-exon AUG circRNAs derived from the *LPAR1* gene (Fig 4A). In the ENCODE data, circLPAR1 is the highest expressed circRNA in 21 samples, and it was the most abundant circRNA in one of the first global analyses of circRNA expression in a human fibroblast cell line, hs68 (Jeck et al, 2013). Here, circLPAR1 was shown to be threefold higher expressed than the second highest circRNA and resistant towards RNase R treatment. We constructed a minigene expression vector including the two exons of *LPAR1* and a portion of the flanking introns (Fig 4B). As with most other AUG circRNA, *LPAR1* has no IAE in close proximity, and the mode of biogenesis is, therefore, currently unclear. To overcome this, we artificially inverted and inserted part of the upstream intron downstream of the splice donor (Fig 4B), which results in clean and efficient

---

relationship between circRNA expression and frequency of AUG-containing circRNA and conservation to mouse, that is, found as circRNA in mouse ENCODE RNAseq. **(D)** Based on the top 1,000 expressed circRNAs, the fraction of circRNA coordinates found as circRNA in the mouse ENCODE data (conserved circRNA) stratified by annotation is shown. Moreover, the fraction of conserved splice sites, that is, corresponding to annotated splice sites in mouse, conserved SD, conserved SA, and sites without annotation are depicted. *P*-value is calculated using Fisher's exact test using conserved circRNAs from AUG and non-AUG stratification. **(E, F)** IAE distance (E) and flanking intron length (F) for circRNAs stratified by annotation or by conservation, and host gene exons. For flanking intron length (F), only circRNAs and exons with annotated up- and downstream introns were included in the analysis. *P*-values are based on Wilcoxon rank-sum tests, and where "AUG circRNAs" are compared with "Other circRNAs" and "Ambiguous circRNAs," only the highest obtained *P*-value is denoted.

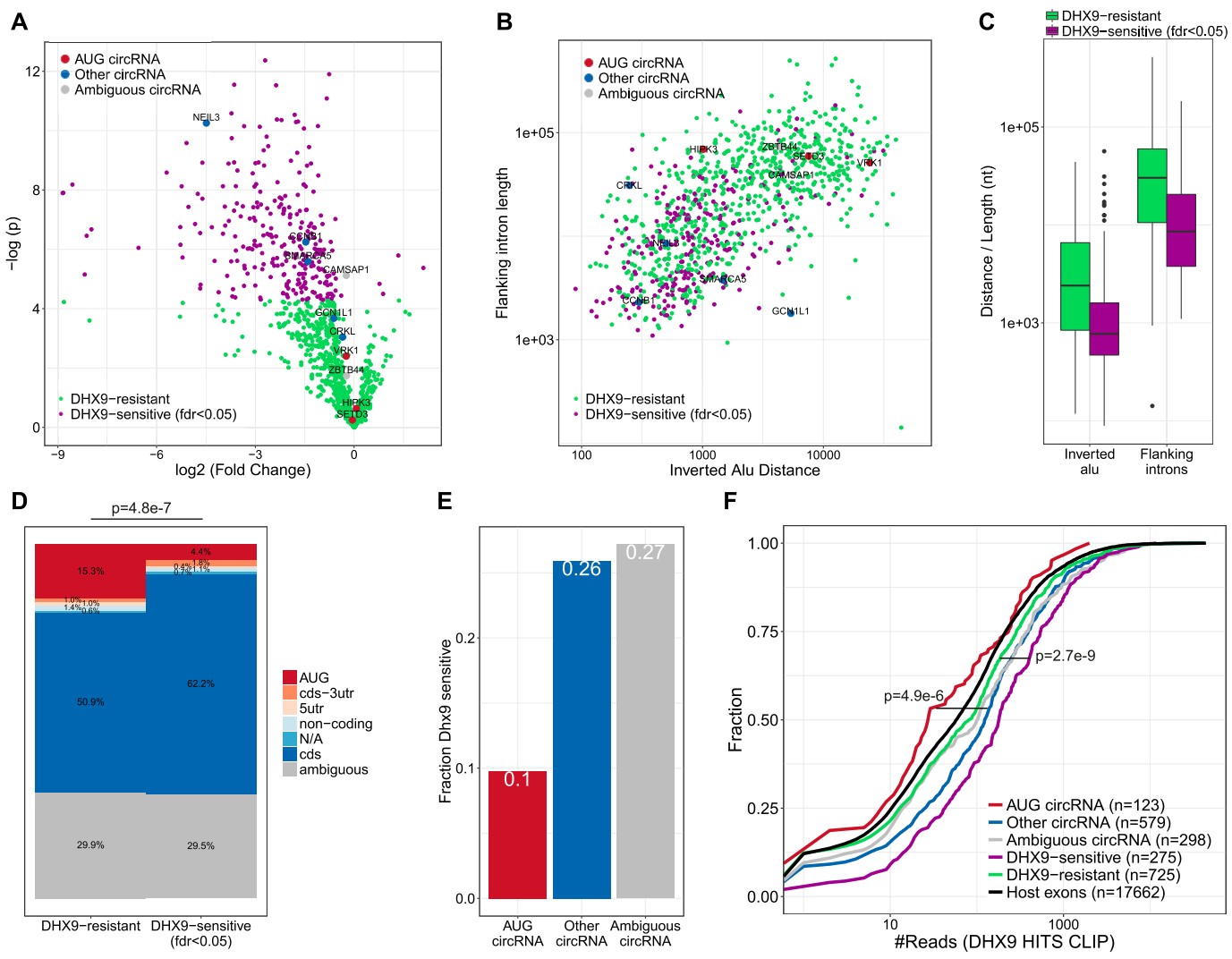

**Figure 3. AUG circRNAs are DHX9-resistant.**
**(A)** Volcano plot on top 1,000 expressed circRNAs from Aktas et al (2017), showing circRNA deregulation upon DHX9 knockdown color-coded by fold-change significance (fdr < 0.05). The top 10 expressed circRNAs are highlighted and color-coded by annotation. **(B)** Scatterplot showing flanking intron length by flanking *Alu* distance. Here, as in (A), the top 10 expressed circRNAs are highlighted. **(C)** Boxplot showing the distribution of IAE distance and flanking intron length for circRNAs stratified by DHX9 sensitivity. **(D)** Frequency of circRNA annotations for circRNAs resistant or sensitive towards DHX9 knockdown. *P*-value is calculated by Fisher's exact test using AUG and non-AUG stratification. **(E)** Within the top 1,000 expressed circRNA, the fraction of DHX9-sensitive species are grouped by annotation and plotted. **(F)** Cumulative plot showing the number of DHX9 HITS-CLIP reads in the flanking vicinity (within 1 kb upstream and downstream of the SA and SD, respectively) of circRNAs. Here, the circRNAs were stratified by either genic annotation or DHX9 sensitivity. *P*-values are calculated by Wilcoxon rank-sum test, and refer to the "AUG circRNA" versus "Other circRNA" and "DHX9 sensitive" versus "DHX9 resistant" subgroups.

expression of circLPAR1 exhibiting RNase R resistance (Fig 4C and D). The protein-coding ability of circLPAR1 was determined by the insertion of an eGFP tag just upstream of the stop codon in the putative ORF (Fig 4B). Here, detection of the tag would only be possible if translation proceeds across the BSJ. First, we tested whether the insertion of eGFP would impede or alter the circularization of circLPAR1. As expected, the circLPAR1-eGFP shows changed migration but remains RNase R resistant (Fig 4D). Then, we over-expressed the untagged and GFP-tagged variants of circLPAR1 in HEK293 cells and performed fluorescent microscopy and western blot analyses; however, in both cases, we were unable to obtain any signal (Fig 4E and F). Based on AUG circRNA, circSLC8A1, and the

ambiguous AUG circRNA, circCDYL, similar vector designs were constructed (Figs S11A and B, S12A and B) and effective circRNA production was observed (Figs S11C–E and S12C–E); however, once again no GFP-positive signal by western blotting or fluorescent microscopy was obtained (Figs S11F and G, S12F and G). Collectively, this suggests that these specific circRNAs are not subjected to translation under normal conditions in HEK293 cells.

In a parallel experiment using the large T antigen transformed HEK293T cells, which normally show higher expression of ectopic transgenes, we surprisingly observed a faint GFP-positive band on the western and few GFP positive cells when overexpressing circLPAR1-eGFP. However, this was seen both for LPAR1 vectors with

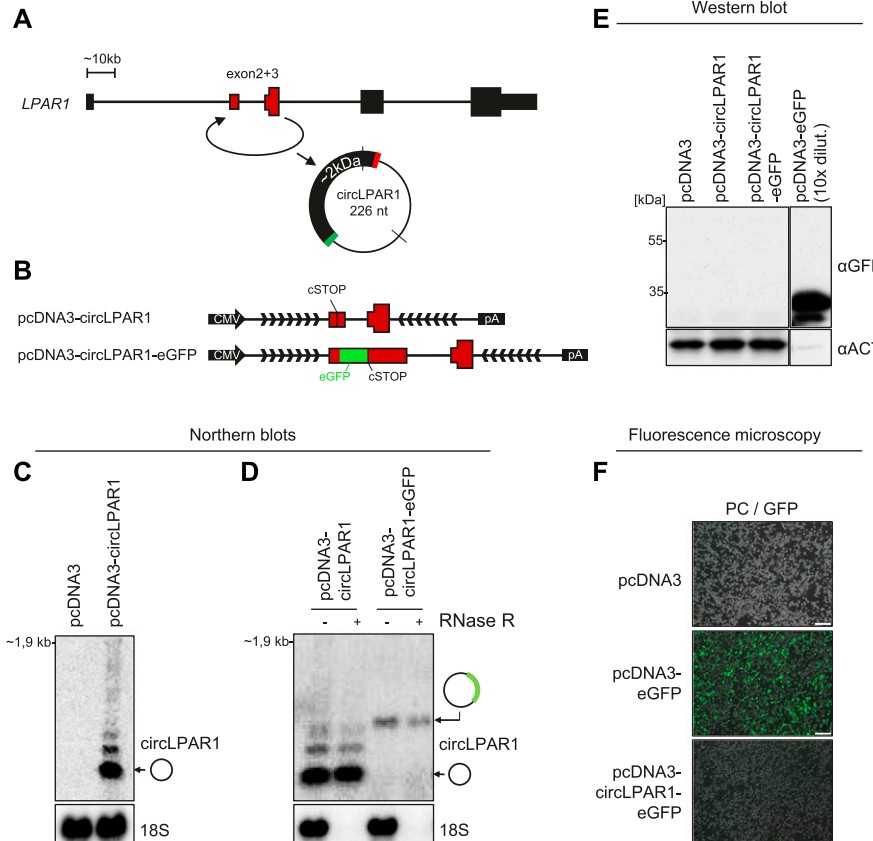

**Figure 4. No evidence of circLPAR1 translation.**
**(A)** Genomic representation of the *LPAR1* host gene locus. The exons are not drawn to scale. **(B)** Schematic representation of expression vectors comprising the CMV promoter, exons 2 and 3 known to circularize, the putative circRNA-specific stop codon (cSTOP), the insertion of eGFP ORF, the flanking regions (divergent arrows indicate artificially introduced inverted element), and the BGH pA signal. **(C, D)** Northern blot analysis of total RNA from ectopic overexpression of circLPAR1 vectors as denoted in HEK293 cells (C) or RNA with or without RNase R treatment (D). The membranes were probed for circLPAR1 as denoted to the right (top panels), and 18S serves as loading and RNase R control (bottom panels). **(E)** Western blot showing GFP expression in HEK293 cells transfected with positive control (pcDNA–eGFP) or circLPAR1–eGFP fusion. **(F)** Merged phase contrast and GFP fluorescence images (PC/GFP) obtained from HEK293 cells transfected with vectors as denoted (scale bars, 200 μm).

or without artificial inverted elements and thus irrespective of circularization (Fig S13A–E). Moreover, vector linearization before transfection reduced the GFP output, although this was also observed for the canonical GFP expression vector. Collectively, this indicates that in extreme conditions, exon repeats are likely produced from "rolling circle" read-through transcription on plasmid templates (see schematics in Fig S13F), and, presumably, this is particularly prevalent in circRNA expression vectors, as the vector-encoded poly(A) signal is situated downstream the SD and, therefore, subjected to U1-mediated repression (Kaida et al, 2010). Thus, vector-based overexpression may generate false-positive protein products from capped mRNA indistinguishable from the predicted circRNA-derived peptides, and conclusions based on ectopic expression setups should be drawn with utmost caution.

## Putative ORFs are not conserved features in AUG circRNAs

As shown above, AUG circRNAs are generally more conserved across species than other circRNAs (i.e., more often found in mouse as a circRNA). If the functional relevance of these AUG circRNAs is to encode protein, features specific for translation should also exert increased conservational restraint. We focused this analysis on the AUG circRNAs within the top 1,000 expressed circRNAs from the ENCODE analysis and used the AUG-containing exon from "other circRNA"-associated host genes (termed the "AUG exon") as exons

with comparable expression level but without any evidence of circularization as a control.

In theory, the circular topology allows for infinite ORFs without stop codons. However, this is only predicted for a very small subset of AUG circRNAs (8% in both human and mouse, Fig S14A and B). For the remaining circRNAs, the predicted ORF terminates shortly after the BSJ (median length of 10 aa after BSJ, Fig S14C and D), which is very close to the expected geometric distribution of stop-codon frequency considering the overall 5'UTR nucleotide composition (Fig S14E and F). This suggests that the predicted lengths of the circRNA derived peptides are very close to what would be expected by chance.

The mRNA ORFs are typically highly conserved between species. In contrast, the 5'UTRs generally exhibit much lower evolutionary constraints. We compared the overall conservation of 5'UTRs but only considering the AUG-containing exon (see schematics in Fig 5A, upper panel). Here, AUG circRNAs show a significantly higher cross-species conservation compared with the control AUG exons (Fig 5A). Next, to elucidate whether the increased conservation coincides with a putative ORF from the annotated AUG across the BSJ and into the 5'UTR, we determined the relative conservation of predicted stop codons. As a positive control, we included the annotated stop codon from the circRNA-derived host genes (termed "Host mRNA") in the analysis. Based on phastCons scores obtained from the UCSC genome browser, the relative conservation of stop codon versus downstream triplet was plotted (Fig 5B). This shows, as expected, a

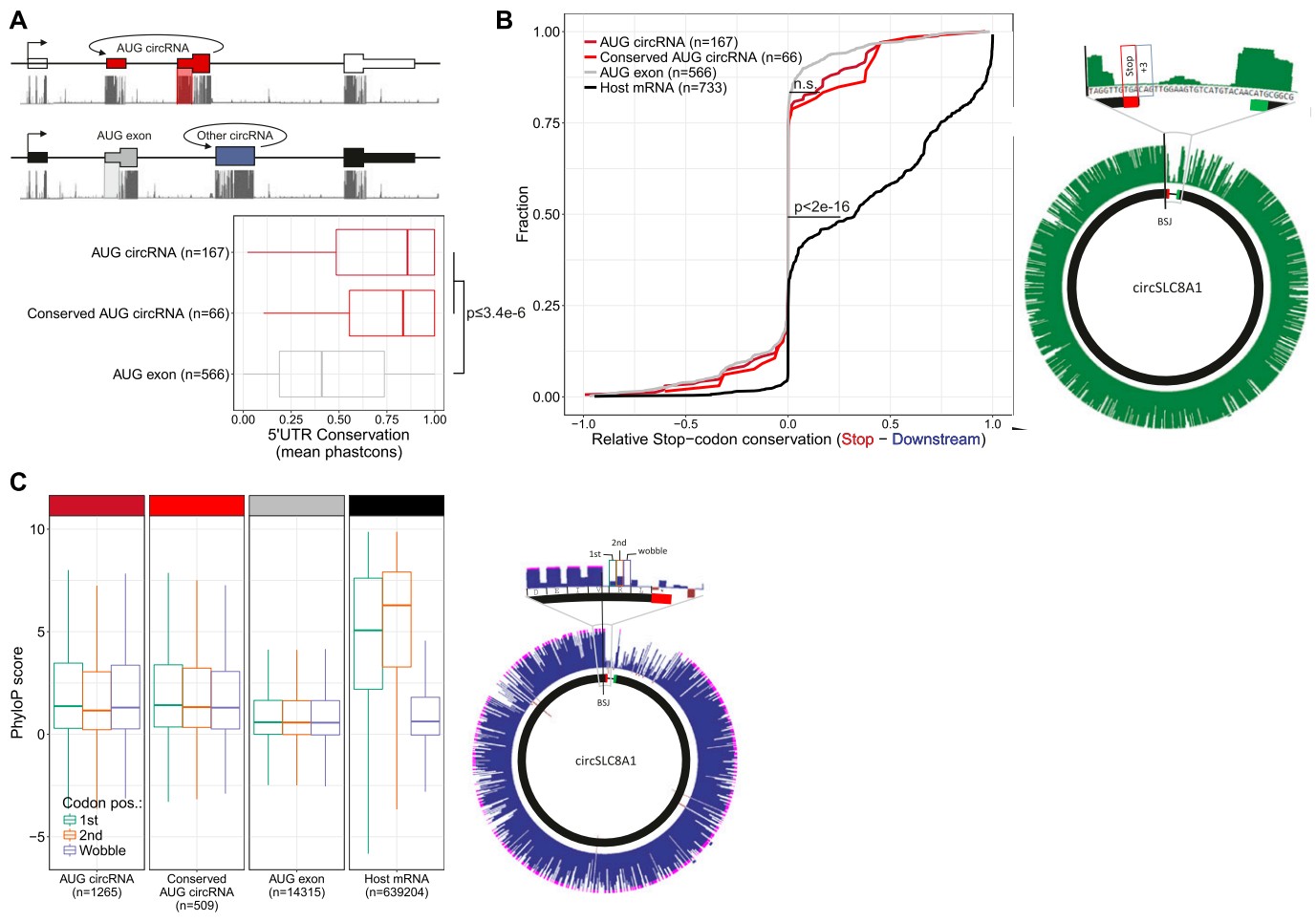

**Figure 5. No evolutionary preservation of ORF in AUG circRNAs.**
**(A)** PhastCons analysis of 5'UTRs within the AUG-containing exon performed on AUG-containing exons in conserved and non-conserved AUG circRNAs, as well as AUG-containing exons from noncircular AUG exons. The 5'UTRs (transparent red and transparent grey) and representative phastCons tracks are depicted for "AUG circRNAs" and "AUG exons," respectively, in the above schematics. **(B)** PhastCons scores of putative circRNA-derived stop codon positions were extracted and compared with the phastCons scores of the immediate downstream triplet (+3) to obtain a measure of selective conservation (relative stop codon conservation). The analysis was stratified by annotation as in (A) and visualized as a cumulative fraction plot. The stop codon and +3 triplet are exemplified by circSLC8A1 to the right. Similar analysis on bona fide stop codons within host gene ORFs is included (black line). **(C)** PhyloP analysis of single-position conservation for first, second, and wobble-position for bona fide ORFs within host genes and putative ORFs after BSJ as exemplified to the right by circSLC8A1. N denotes number of codons analyzed.

notable conservational enrichment of host mRNA stop codons; however, no significant difference between AUG circRNAs and AUG exons is observed, and not even when focusing the analysis on the conserved subset of AUG circRNAs. In agreement with our analysis of ORF lengths, this suggests that the putative stop-codon sequence is not under evolutionary constraints. Finally, we determined the third-nucleotide (the wobble) conservation relative to the two other nucleotides in every codon within the putative ORF after the BSJ (Fig 5C, schematic). Based on phyloP basewise conservation, the annotated mRNA ORFs show a clear and significant decrease in wobble nucleotide conservation, in accordance with previous analyses (Chamary et al, 2006). However, for the AUG circRNA, again, no differential conservation between the wobble position and the two other bases is observed, not even for the conserved subset of AUG circRNA (Fig 5C), supporting the preliminary conclusion that the coding properties of AUG circRNAs are

not conserved. Consistently, similar analyses on the murine repertoire of AUG circRNAs produce almost identical results (Fig S15A–C), suggesting little or no preservation of the circRNA-specific ORF; however, the exact peptide sequence encoded after the BSJ could be of less significance and, therefore, not under evolutionary pressure, but still the protein output could be functionally important.

## AUG circRNAs are generally not templates for translation

To assess the translational potential of circRNA globally, we took advantage of the wide range of ribosome profiling (RiboSeq) data currently available online: ~500 and ~1,300 samples from human and mouse origin (Table S5). After adapter trimming, we obtained a total of approximately 22 and 24 billion reads between 25 and 35 nucleotides in length from human and mouse, respectively,

consistent with an approximate ribosome footprint of ~30 nts (Ingolia et al, 2009).

In general, the distance from the 5' end of footprinting reads to the ribosome P-site (P-site offset) is 12 nts (Ingolia et al, 2009; Bazzini et al, 2014); however, for shorter and longer reads this offset varies (Dunn & Weissman, 2016). Moreover, we noted that there was a large degree of P-site offset variation between samples, and consequently, we initially analyzed each sample individually by mapping all the 25–35-nt reads onto the *GAPDH* and *ACTB* (*β*-actin) mRNAs. For each read length, we determined the amount of on-frame P-sites for 12, 13, and 14 nts offsets and the associated *P*-value by binomial test (see example in Fig S16), which was used as a measure of dataset fidelity. Thus, in all samples, the efficiency by which each read length (25–35 nts) is able to demarcate translation from noise was determined (Figs S17 and S18). Here, for both species, the 28–31-nt reads show the highest abundance and fidelity (Fig S19A and B).

We then applied the reads to circRNAs using the 5' offset with the lowest *P*-value in the above analysis. To evaluate translation of circRNA, the BSJ is the only circRNA-specific sequence. Therefore, we concatenated the circRNA exons on all the top 1,000 expressed circRNAs from the ENCODE data to display the BSJ in a linear manner compatible with short read mapping. As above, we also included the AUG-containing exon from "other circRNA"–derived host genes ("AUG exon"), as well as "ambiguous AUG circRNAs", that is, circRNAs derived from ambiguous host gene isoforms of which at least one is annotated to contain the AUG start site (see schematics on Fig 6A). By plotting the distribution of reads across the BSJ, only a small fraction of reads spans the BSJ compared with the immediate upstream regions (Fig 6B). However, in contrast to previous reports (Guo et al, 2014; You et al, 2015), there is a notable fraction of BSJ-spanning reads defined here as P-site position from –8 to +6 relative to the BSJ (Fig 6B) comprising a 15-nt stretch (5 codons). To ensure that BSJ-spanning reads are in fact likely derivatives of circRNAs, the reads were re-aligned to an assembled transcriptome allowing one mismatch. Particularly for the human RiboSeq data, this discards most reads spanning the BSJ on the "AUG Exon" subset. In contrast, almost all reads mapping perfectly to the BSJ of bona fide circRNA have no detectable mRNA alignment (Fig S20A and B). Now, when considering the likelihood of each read to actually derive from translating ribosomes, only ~20% of BSJ-spanning reads are from high quality samples (with fdr < 0.01), whereas ~70% of the upstream-derived reads are of high quality (Fig 6C). This suggests that across the BSJ, the quality of ribosome profiling data is of particular high relevance and that noise consumes most of the BSJ-spanning reads. This difference in quality is corroborated in mouse although here the difference in quality is less pronounced (Fig S21A–C). Nonetheless, to address translation across the BSJ, we filtered out the low-quality reads (fdr > 0.01) and used the remaining reads to determine whether phasing in accordance with translation of the putative ORFs is evident. Here, we simply counted the number of reads in-frame and out-of-frame on the 5 codon BSJ-spanning stretch and compared this to a 5 codon stretch immediately upstream the BSJ. For all subtypes of circRNA ("AUG circRNA," "Ambiguous AUG circRNA," and "Other circRNA"), a roughly equal distribution of reads between all three frames is observed across the BSJ, whereas for the upstream region, approximately 50–60% of

reads are in-frame, both for humans and mouse (Figs 6D and S21D). This strongly suggests that the AUG circRNAs as a whole, or any of the other circRNAs for that matter, are not subjected to translation as evidenced by RiboSeq analysis. However, it is likely that a small and restricted subset of circRNAs is acting as templates for translation, and, therefore, the signal from these drown in the noise from others. To evaluate this, we analyzed all top 1,000 circRNAs with at least 10 BSJ-spanning RiboSeq reads individually. Here, in humans only circUBXN7—an ambiguous AUG circRNA for which the annotated host gene has multiple start codons—shows an enrichment of in-frame reads although not significant when evaluating unique reads only (*P*[unique reads] = 0.10, one-tailed binomial test, Fig 6E, unique reads shown in parentheses). Similarly, in mouse, no significant phasing of unique reads is observed across the circRNA BSJ (Fig S21E).

## No circRNA-specific peptides are found in proteomics data

To strengthen the conclusion drawn from the previous analyses, we conducted an extensive mining of proteomics data. Based on the top 1,000 expressed circRNAs derived from protein-coding host genes, we predicted the resulting peptide from each circRNA, and compiled an associated database of decoys (Fig 7A). Using the established mass-spec search tool, comet (Eng et al, 2013), we searched through 35 mio. spectral data (see Table S6) derived from the nci60 cell lines, the human proteome draft map, as well as the human brain proteome. All hits and the corresponding e-values were then assigned to either host gene, circRNA, or decoy. Here, in contrast to host genes, the e-values of circRNA-derived peptides are not significantly different from the decoy peptides (Figs 7B and S22A). In fact, using a relaxed fdr cutoff of 0.2, not a single circRNA-derived peptide is identified (Fig 7C), and the AUC in a ROC analysis was for all circRNA subgroups close to 0.5 (Fig S22B). This adheres with the above conclusions that circRNAs are unlikely to be substrates for translation. Moreover, we focused specifically on circRNAs shown to produce protein in the literature and subjected these candidates to both RiboSeq and mass spectroscopy analysis (summarized in Table S7). Here, in all cases, no supportive RiboSeq reads are found, and not a single circRNA-specific peptide with e-values below 0.05 is observed. In fact, for many of the putative peptides derived from circRNAs in Yang et al (2017), the associated ORFs have no canonical start codon or the circRNAs are without any reads in the ENCODE RNAseq. This indicates to us that many of these identified peptides are likely false positives.

## CircRNAs are not enriched with 8mer miRNA target sites

Finally, to determine whether AUG circRNAs show evidence of miRNA *sponging*, we characterized the frequency of miRNA target sites on circRNAs. Consistent with other analyses, we focused on the top 1,000 expressed circRNA with splice sites derived from protein-coding genes (i.e., ciRS-7 is not included in this analysis), the AUG exons from "other circRNA"–producing host genes, and the unique 3'UTRs from the top 1,000 expressed circRNA host genes. We conducted an 8mer target site analysis based on all the conserved and confident-assigned miRNAs from miRbase (n = 338). Here, we observe for all target categories that the overall numbers of 8mers

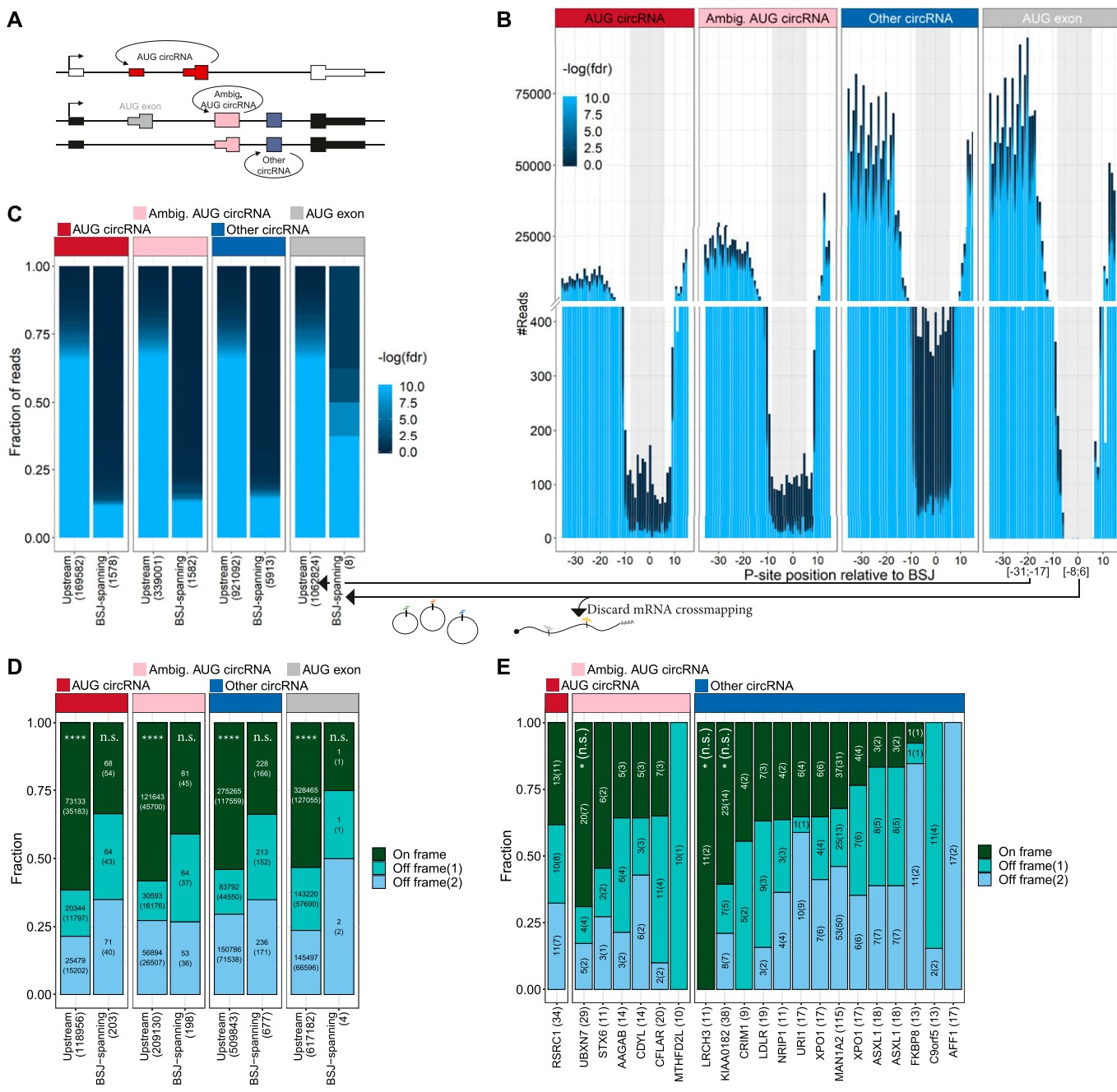

**Figure 6. Ribosome profiling reads across BSJ.**
**(A)** Schematics showing circRNA annotation of "AUG circRNA," "Ambiguous circRNA," and "Other circRNA." **(B)** Based on ribosome profiling datasets, the number of ribosome P-sites around the BSJ were counted for each subclass of circRNA ("AUG circRNA," "Ambiguous AUG circRNAs," and "Other circRNA," see text for more detail). The AUG-containing exon from non-AUG ("AUG exon") circRNA host genes. The plot is color-scaled according to the associated read-class P-value (See Figs S16–S18). The grey box denotes the defined P-site position of BSJ-spanning reads, from pos –8 to +6 relative to the BSJ. **(C)** Based on all BSJ-spanning reads (P-sites from –8 to +6 relative to BSJ) and upstream reads (–31 to –17 relative to BSJ), the fdr-value distribution based on RiboSeq quality assessment is shown. **(D)** Phasing of reads across BSJ. Here, based solely on high-quality reads (fdr < 0.01), the fraction of P-sites in-frame and out-of-frame across the BSJ (–8 to +6) are shown for each subclass of circRNA. The number below the plot represent total number of reads analyzed, whereas the numbers inside the plot reflect total or unique (in parenthesis) counts within each frame (****$P <$ 0.0001). **(E)** As in (D), but for each individual circRNA with 10+ reads across the BSJ (*$P <$ 0.05). P-values in (D) and (E) represent bonferroni-corrected one-tailed binomial tests.

are enriched compared with expected based on permutations (Fig S23A). However, this is also observed for the reverse complement sequence, suggesting that mere nucleotide frequency is not a suitable null hypothesis for in silico target site prediction. Instead, considering the phastCons scores of all predicted target sites, we observe, as expected, that the 3'UTR targets are significantly more

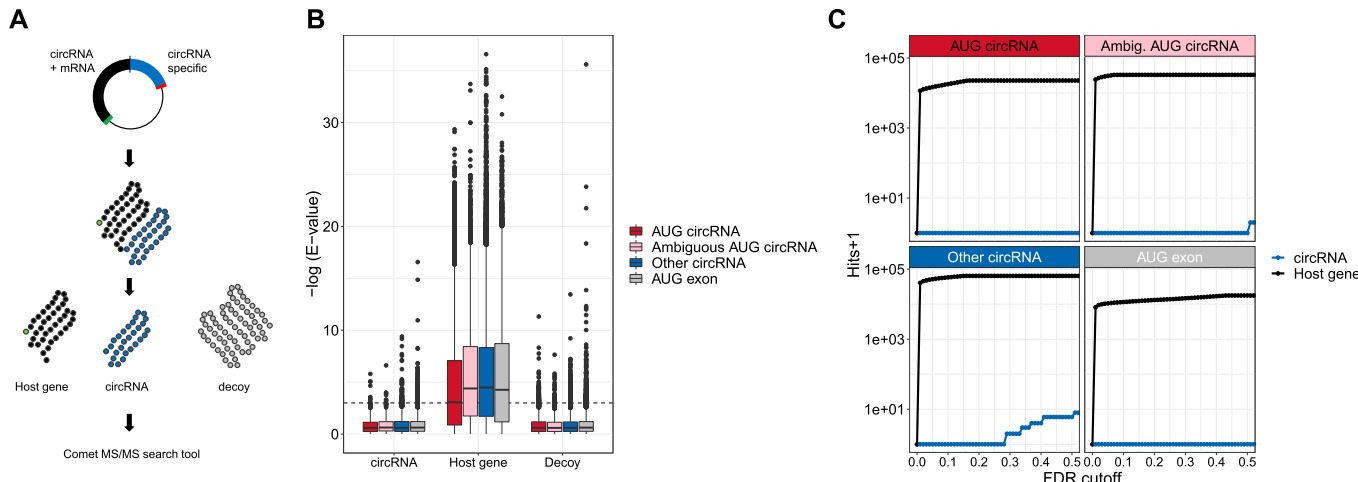

**Figure 7. circRNA-derived peptides.**
**(A)** Schematics outlining the proteomics analysis pipeline. For the top 1,000 expressed circRNAs, the putative ORFs were deduced. From this, a reference proteome was established comprising the entire circRNA-derived peptides and a matched list of decoys. Then, proteomics data were queried against the reference database using comet and hits were stratified by "circRNA," "host gene," and "decoy." **(B)** Boxplot showing the distribution of obtained e-values from "circRNA," "host gene," or "decoy" subclassified into circRNA annotation as shown. **(C)** Cumulative sum of peptide hits with incrementing false discovery rate (FDR) cutoffs. FDRs are calculated as the cumulative fraction of decoys on e-value–sorted data with circRNAs and host genes analyzed separately.

conserved compared with the reverse complement and the permutated sequences (Fig S23B). This, in fact, is also observed for AUG exon, that is, noncircularized exons with annotated start codons, and to a lesser degree, the AUG circRNAs. This could indicate that preserved miRNA targeting in 5′UTRs is occurring, but not in a circRNA-specific manner. To elucidate any putative circRNA sponges, we counted the frequency of all circRNA:miRNA pairs. Here, no circRNAs were observed with more than three target sites for any particular miRNA, in contrast to ciRS-7 with 59 8mer sites for miR-7. In one outlier instance, however, hsa-miR-342-5p has 23 distinct target sites in the 3′UTR of ANKRD11 (uc002fnf.1) (Fig S23C). As a last resort, based on the Smith–Waterman alignment score for each putative target site, we sought to disclose whether circRNAs were more prone to miRNA-mediated cleavage than expected by change; however, here no significant differences were observed between the actual target sequences and the controls (Fig S23D). Conclusively, this aligns with previous studies showing no enrichment of miRNA target sites on circRNAs—except for ciRS-7 (Guo et al, 2014) and suggests that circRNAs are unlikely to serve as miRNA sponges.

## Discussion and Conclusion

Here, by thorough disclosure of the circRNA landscape across human and murine tissues and cell lines, we have shown that a certain subclass of circRNAs, namely, the AUG circRNAs, are abundantly expressed and conserved across species. The high abundance of AUG circRNAs is partly reflected in increased circular-to-linear ratio, suggesting increased backsplicing efficiency. In fact, for the top 1,000 expressed circRNAs, there is no significant difference in linear spliced reads between AUG circRNA and Other

circRNAs. This indicates that the cross-species conservation is not merely a consequence of abundant host gene expression but may instead reflect sequence preservation of the AUG circRNAs specifically. Moreover, the AUG circRNAs associate with very long flanking introns and are devoid of flanking IAEs in contrast to most other circRNAs. The *Alu*-independent biogenesis is consolidated by analysis of RNAseq on DHX9 depleted cells and DHX9 HITS-CLIP, where AUG circRNAs generally are insensitive to DHX9 perturbation and exhibit reduced DHX9 binding in the flanking regions compared with other circRNAs. This strongly suggests that AUG circRNAs are not aberrant RNA species occasionally escaping the DHX9 mediated surveillance. CircHIPK3, the most abundant circRNA in humans, has proximal IAE and is considered an *Alu*-dependent circRNA. Our analysis shows that circHIPK3 escape regulation by DHX9 suggesting *Alu*-independent biogenesis. Moreover, circHipk3 is also highly expressed in mouse, but here devoid of any inverted *Alu* elements in the flanking regions. Instead, the mouse and human loci share the long flanking introns, suggesting that the biogenesis of circHIPK3 may not rely solely on IAEs in the flanking regions. Here, the combination of long flanking introns and proximal IAE could in part explain the high overall circHIPK3 expression levels in cells and tissue. Thus, we conclude that most AUG circRNAs use *Alu*-independent routes of biogenesis that most likely depend on the length and features of the flanking introns. In *Drosophila*, a well-established circRNA from the laccase locus is flanked by inverted repeats required for biogenesis (Kramer et al, 2015); however, consistent with our results, circRNAs from *Drosophila* associate in general with very long introns but without any obvious flanking sequences engaging in intron–intron base-pairing (Westholm et al, 2014). Instead, splicing kinetics and backsplicing seem to be closely related features. It has been shown that increased polymerase speed results in decelerated linear splicing (Carrillo Oesterreich et al, 2016) but stimulates backsplicing

(Zhang et al, 2016b). Similarly, depletion of splicing factors increases circRNA production by affecting splicing kinetics negatively (Liang et al, 2017). Therefore, we propose that some embodiment of slow splicing kinetics—which may be commonly associated with long introns—is a prerequisite for the production of many circRNAs. Whereas in higher eukaryotes, the duration of splicing does not generally correlate with intron length (Hollander et al, 2016), the duration of intron transcription and the concomitant delay of splicing could in itself be of importance. Whether the intrinsic features of long intron splicing are sufficient for efficient backsplicing or whether certain trans-acting factors are required as well is currently unknown, and as such, the exact production of the most abundant and conserved circRNAs remains elusive.

Based on cross-species sequence analysis, the 5′UTR sequences contained within AUG circRNAs are overall more conserved than other noncircular 5′UTR elements. However, based on single nucleotide constraints, this conservation is not due to circRNA-specific ORFs. Consistently, upon overexpression of three different AUG circRNAs, no circRNA-specific peptides were detected. In addition, thorough and extensive RiboSeq analysis suggests that BSJ-spanning reads are not derived from translating ribosomes assuming that the P-site offsets deduced from house-keeping genes (*GAPDH* and *ACTB*) also apply to circRNA translation. Finally, analysis of 35 mio. peptide spectra from mass spectrometry fails to identify a single circRNA-derived peptide with a fdr below 0.2. Collectively, this argues that circRNAs in general and AUG circRNAs specifically are not subjected to translation. It should be emphasized that our analyses solely focus on human and mouse samples and, therefore, the functional features of circRNA in other organisms, such as *Drosophila*, has not been addressed here. Moreover, our results do not necessarily exclude inefficient translation of circRNAs or restricted translation solely under specific conditions. Consequently, it is possible that published circRNAs, such as circMbl in *Drosophila* (Pamudurti et al, 2017), and circZNF609, circSHPRH, circFBXW7, and circLINC-PINT in human cells (Legnini et al, 2017; Zhang et al 2018a, 2018b; Yang et al, 2018), indeed engage in protein production; however, our data strongly point towards translation of circRNAs as being a rare and uncommon process.

Since the functional characterization of ciRS-7 as a dedicated miR-7 sponge or regulator, many examples of circRNAs acting as miRNA sponges have been proposed. Apart from the 70+ selectively conserved miR-7 sites on ciRS-7, the miRNA sponge potential seems not to be a conserved feature of circRNAs (Guo et al, 2014). Moreover, stoichiometric analysis of circRNA:miRNA:mRNA ratios suggests little or no overall effect upon circRNA-mediated miRNA inactivation (Denzler et al, 2014; Jens & Rajewsky, 2015), highlighting that the notion of circRNAs as miRNA sponges is very controversial. This is consistent with our analysis of putative 8mer target sites. Here, the circRNAs showed no overall enrichment of target site frequency, target site conservation, target site alignment scores, nor any emerging circRNA:miRNA pairs, suggesting that from a bioinformatics point of view, circRNAs do not classify as miRNA regulators.

Instead, specifically for the AUG circRNAs, expressing translationally inert canonical start codons in its natural sequence context

could be useful in certain scenarios, for example, as binding platforms for regulators of translation factors in the cytoplasm, and using the circular topology for this purpose seems plausible, although the same stoichiometric issues as for the miRNA sponge hypothesis may apply here. It is possible that the functional relevance of one particular AUG circRNA is very subtle, but that the accumulated contribution of all the circRNAs are of physiological importance. As such, the highly stable and durable circRNAs could constitute a background of nonresponsive RNA entities in the cell to ensure robustness by transiently associating with RNA-binding proteins in the cytoplasm, thereby reducing nonspecific and potentially detrimental RNA–protein interactions. In any case, future research will undoubtedly shed light on the elusive mechanism by which these highly abundant circRNA species are produced and more interestingly elucidate the functional capabilities of AUG circRNAs.

# Materials and Methods

### Plasmids

All plasmids were generated by PCR with subsequent restriction digest and ligation into pcDNA3 (Invitrogen). Primers are listed in Table S8.

### Cell lines and transfection

HEK293 Flp-In T-Rex cells (Invitrogen) or HEK293T cells (ATCC) were used for all experiments. The cells were cultured in DMEM with GlutaMAX (Thermo Fischer Scientific) supplemented with 10% FBS and 1% penicillin/streptomycin sulphate. The cells were kept at 37°C and 5% $CO_2$. Transient transfections were carried out using calcium phosphate as transfection reagent using standard procedures or Lipofectamine Reagent 2000 (Invitrogen) accordingly to the manufacturer's protocol. After 24 h, the medium was changed and 48 h post transfection cells were harvested either by resuspension in (i) 2× SDS loading buffer (for Western blotting) or (ii) TRIzol Reagent (Thermo Fisher Scientific) (for RNA purification) adhering to the manufacturer's protocol (see below). For linearization of plasmids before transfections, the respective plasmids were digested with FastDigest PvuI (Thermo Fisher Scientific).

### Northern blotting

Northern blotting was performed as described by Hansen, 2018a. Briefly, 10 $\mu l$ RNA (1g/l) and 20 $\mu l$ Northern loading buffer (58.8% formamide, 6.5% formaldehyde, ethidium bromide, 1.18% MOPS, and bromophenol blue) were mixed and denatured at 65°C for 5 min. The RNA was separated by electrophoresis on a 1.2% agarose gel containing 3% formaldehyde and 1× MOPS at 75 V. After electrophoresis, the gel was briefly washed in water and exposed to UV to visualize the EtBr stained ribosomal RNA bands. The gel was transferred to a Hybond N+ membrane (GE Healthcare) overnight (O/N) in 10× SSC. Then, the membrane was UV cross-linked and pre-hybridized in Church buffer (0.158 M $NaH_2PO_4$, 0.342 M $Na_2HPO_4$, 7%

SDS, 1 mM EDTA, and 0.5% BSA, pH 7.5) for 1 h at 55°C and subsequently probed with a 5′ radioactively labelled DNA oligonucleotide (see Table S8) at 55°C O/N. The next day, the membrane was washed twice in 2× SSC, 0.1% SDS for 5 min and twice in 0.2× SSC and 0.1% SDS for 15 min. All washes were carried out at 50°C. Finally, the membrane was exposed on a phosphoimager screen and analyzed using Quantity One or Image Lab software (Bio-Rad). The membranes were stripped in boiling stripping buffer (0.1% SDS and 1 mM EDTA).

### RNase R

For RNase R experiments, 4 µg RNA was digested with 4 U RNase R (Epicentre) in a total reaction volume of 10 µl for 10 min at 37°C. Then, 20 µl Northern loading buffer was added and heated at 65°C for 5 min before loading on an agarose northern gel. (see above).

### Western blotting

Cells were harvested in 1× PBS and centrifuged at 150 $g$ at 4°C for 5 min. The supernatant was removed and the cell pellet was lysed directly and resuspended in 2× SDS loading buffer (125 mM Tris–HCl, pH 6.8, 20% glycerol, 5% SDS, and 0.2 M DTT). After resuspension, the samples were boiled at 95°C for 5 min before loading 1% on a 10% Tris-glycine SDS–PAGE gel with 10 µl PageRuler Plus Prestained Protein Ladder (Thermo Fisher Scientific). After ~1 ½ hours, electrophoresis proteins were immobilized on an Immobilon-P Transfer Membrane (EMD Millipore) overnight in a wet-blotting chamber. The next day, the membrane was pre-incubated at RT with 20% skim milk to block unspecific binding for 1 h. Then, primary antibody (Table S8) in blocking solution was added and incubated 1 h at RT, followed by 1-h incubation with secondary antibody in blocking solution. After each antibody incubation, the membrane was washed 3 × 5 min in 1× PBS with 0.05% Tween 20 and subsequently with 1 × 5-min wash with 1× PBS. Exposure was performed using SuperSignal West Femto Maximum Sensitivity Substrate kit (Thermo Fisher Scientific) and Amersham Hyperfilm ECL (GE Healthcare).

### Microscopy

48 h after transfection, the cells were imaged live using phase-contrast and fluorescence microscopy with normal FITC filter set (using 1-s exposure, ISO200) on an Olympus IX73 microscope. Images were merged using ImageJ.

### RNAseq datasets and circRNA detection

Raw RNA sequencing data were downloaded from the ENCODE Consortium (www.encodeproject.org) or the NCBI Gene Expression Omnibus (www.ncbi.nlm.nih.gov/geo/) (see Tables S1 and S3). CircRNA prediction was performed by find_circ, version 1.0 (Memczak et al, 2013), and circexplorer2.0.1 (Zhang et al, 2016a) adhering to the recommendation by the authors. For find_circ, an increased stringency threshold was used requiring that both adaptor sequences map with highest possible mapping quality (mapq = 40) (Hansen,

2018b). Moreover, for find_circ and CIRCexplorer2, only circRNAs supported by at least two reads in a given sample was kept. CircRNAs found by both algorithms with the abovementioned stringency were used in subsequent analyses. CircRNA expression was based on BSJ-spanning reads according to find_circ quantification. Likewise, the circular-to-linear ratio was determined by the total number of BSJ-spanning reads multiplied by two and divided by the total number of linear spliced reads spanning the upstream and downstream splice sites, respectively, as determined by find_circ. RPM values were calculated for each sample as the number of BSJ-spanning reads divided by the total number of reads. mRNA expression (FPKM values) was quantified using cufflinks (Trapnell et al, 2010).

### circRNA annotation

Annotation of circRNAs was based on UCSC Genes tracks (hg19 and mm10). First, the annotation database was queried for host genes sharing both the circRNA-specific splice sites. If none were found, genes sharing at least one splice site were queried, and as a last resort, genes fully covering the circRNA locus were retrieved. Similarly, host gene exons were retrieved from annotated isoforms sharing both circRNA producing splice sites but omitting first and last exons and the exons with splice sites coinciding with the circRNA, and duplicate exons were discarded. The circRNA subclass, that is, "AUG circRNA," "CDS circRNA," etc., was only determined based on host gene ORF annotation, and if multiple host gene entries were recovered with divergent annotation, the circRNA was categorized as "ambiguous." To guide detection of circRNAs by circExplorer2 and to facilitate proper annotation of known circRNAs, two additional entries were manually added to the gene annotation database (shown in Table S9). Flanking intron lengths were based on the host gene exon–intron structure immediately upstream and downstream the backsplicing spice sites. In case of multiple isoforms with varying intron length, the mean of all flanking introns found was calculated, whereas circRNAs with no flanking introns annotated were discarded. To extract the distance to nearest flanking IAE, the UCSC RepeatMasker tracks (hg19 and mm10) were used. Here, the 20 most proximal but flanking *Alu* elements were retrieved irrespective of intron–exon structure on either side of the circRNA, and based on these, the closest possible inverted pair was determined.

### Conservation of circRNA

Human circRNAs were converted to mouse (mm10) coordinates using the UCSC liftover tool. Based on the predicted circRNAs from ENCODE mouse data, and on the UCSC gene annotation database, the coordinates coinciding perfectly with mouse circRNA or annotated splice sites were assessed. Converted coordinates not found as splice sites and unmapped coordinated were grouped together as "non-conserved."

### HITS-CLIP analysis

For HITS-CLIP analyses, reads were adaptor-trimmed using trim_galore and barcodes were subsequently removed. The reads

were pair-wised mapped on to the human genome (hg19) using bowtie2 using default settings, and mapped reads were extracted using SAMtools. Reads mapping in the flanking introns within 1,000 bp of the backsplicing splice sites of the top 1,000 expressed human circRNAs were counted and compared.

## Conservation of ORF

Conservation analysis was performed for all top 1,000 circRNAs with both splice sites annotated in a protein-coding host gene. For "AUG circRNAs" and "AUG exons," the ORF was predicted based on the annotated start codon whereas for "Other circRNAs," the longest finite ORF traversing the BSJ was used. Infinite ORFs were not considered in this analysis. In case of multiple isoforms, that is, alternative exons within the circRNAs, all isoforms were considered equally in the analysis. The 5′UTR conservation and stop codon conservation were based on the phastCons scores from 100 species (UCSC) using the hg19 or mm10 reference for human and mouse, respectively. For 5′UTRs, only the AUG-containing exon was analyzed, and for stop codons, the stop codon triplet and the immediate downstream triplet irrespective of position was evaluated. For wobble position analysis, only the circRNA-specific ORF within the 5′UTR was analyzed. Here, for each position (first, second and wobble) in each codon, the PhyloP scores from 100 species (UCSC) was retrieved and analyzed. The ORF lengths were based on number of codons from BSJ to stop, and the geometric distribution of stop codon probability considering the frequency of the individual nucleotides in the 5′UTR region was used to determine the expected ORF lengths.

## RiboSeq analysis

Ribosome profiling (RiboSeq) datasets (see Table S5) were trimmed using trim_galore, and only reads between 25 and 35 nucleotides in length were kept and mapped onto *GAPDH* and *ACTB* mRNA (UCSC accessions; uc001qop.2 and uc003sot.4, or uc009dts.2 and uc009ajk.2, for human and mouse, respectively). For each read length in each RiboSeq sample, an offset of 12, 13, or 14 nucleotides was tested to determine the best possible offsetting based on one-tailed binomial tests, for example, how many codons exhibit more on-frame than off-frame reads (see Figs S16–S19). The offset with the lowest mean *P*-value (obtained from *GAPDH* and *ACTB*) was used for each given read length in a given sample. The mature sequence of all top 1,000 circRNAs with both splice sites annotated in a protein-coding host gene were then concatenated to allow mapping across the BSJ. All RiboSeq reads were then bowtie-mapped to the concatenated circRNA sequences allowing no mismatches using the following arguments: *bowtie -S -a -v 0*. The offset from the quality assessment on *GAPDH* and *ACTB* was used to obtain P-site position. P-sites within [-8; 6] relative to the BSJ was defined as BSJ-spanning, whereas P-sites immediately upstream the SD [-31; -17] was defined as linear upstream reads. Putative BSJ-spanning reads were mapped against an mRNA reference (build on UCSC annotations) with one mismatch tolerance, *bowtie -f -v 1*, and omitted from downstream analysis if mapped. The annotated AUG was used to predict the circRNA-specific ORF; however, in "other circRNAs," the longest

possible ORF traversing the BSJ was used. Again, infinite ORFs were not considered. P-site positions relative to the predicted frames were counted and analyzed. The statistical significance of the proportion of in-frame reads was determined by one-tailed binomial test with 1/3 probability of in-frame reads.

## Mass-spec analysis

First, a reference proteome database was compiled. This contained all well-known contaminants, the UniProt protein collection (June 27, 2018, n = 73,099), and all the putative proteins produced from the circRNAs subjected to RiboSeq analysis. Then, decoys were generated from all the circRNA-derived proteins using the decoyPyRat python script (https://www.sanger.ac.uk/science/tools/decoypyrat) and concatenated to the proteome reference database. Raw MS files (see Table S6) were converted to mgf format using MSConvert (Chambers et al, 2012). Then, the spectra were searched against the reference database using the comet search engine (Eng et al, 2013) adhering to the suggested fixed and variable modifications for each sample. Only the peptides derived from circRNA-based ORFs and the decoys were analyzed, and the peptides were stratified by overlap with the UniProt assembly indicating host gene or circRNA origin. Scores and e-values were extracted and analyzed. Fdr values were calculated as the cumulative fraction of decoys ordered by e-value for circRNAs and host genes, respectively.

## circRNA sponge analysis

To determine miRNA target sites on circRNAs, all "confident" human mature miRNAs were retrieved from miRBase version 21. Only miRNAs with phastCons scores > 0.95 were kept (n = 338). Then, 8mer seed matches (using miRNA positions 2–8 and a downstream A) were searched in the sequence from the top 1,000 expressed human circRNAs with both splice-sites annotated in protein-coding host genes. In addition, all unique 3′UTRs from circRNA-producing host genes were included. For each analyzed target sequence, the reverse complement sequence and 20 random permutations were included in the analysis. The number of putative targets identified were normalized to the expected number of targets based on the nucleotide compositions of target sequence and miRNA seed-match. Conservation of target sites where based on the phastCons score of the seed-match. Smith–Waterman alignments (using 2, –1, and –1 scores for match, mismatch, and gap, respectively) were generated between the full miRNA sequence, excluding the first nucleotide, and the 8mer target site, including an additional 18 nucleotides upstream. Finally, for each target species, all circRNA: miRNA pairs were grouped by counts and summarized.

## Statistical analyses

All statistical analyses are based on Wilcoxon rank-sum tests except if explicitly noted otherwise. Fdr values reflect Benjamini–Hochberg–adjusted *P*-values, except for the mass-spec analysis (see above).

# Supplementary Information

# Acknowledgements

We would like to thank Karoline Krogh Ebbesen for the insightful discussions and critical reading of the manuscript. This work was supported by the Novo Nordisk Foundation (NNF16OC0019874 to TB Hansen).

## Author Contributions

LVW Stagsted: investigation, visualization, and writing—original draft, review, and editing.
KM Nielsen: investigation.
I Daugaard: investigation.
TB Hansen: conceptualization, data curation, formal analysis, supervision, funding acquisition, visualization, and writing—original draft, review, and editing.

## Conflict of Interest Statement

The authors declare that they have no conflict of interest.

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
