## [Reviewer comments · Life Science Alliance]

Life Science Alliance

Non-coding AUG circRNAs constitute an abundant and conserved subclass of circles

Author information redacted

DOI: <https://doi.org/10.26508/lsa.201900398>

Corresponding author(s): *Thomas Hansen, Aarhus University*

Review Timeline:	Submission Date:	2019-04-11
	Editorial Decision:	2019-04-11
	Revision Received:	2019-04-14
	Accepted:	2019-04-16

Scientific Editor: Editor Life Science Alliance

Transaction Report:

Please note that the manuscript was previously reviewed at another journal and the reports were taken into account in the decision-making process at Life Science Alliance. Since the original reviews are not subject to Life Science Alliance's transparent review process policy, the reports and author response cannot be published.

April 11, 2019

RE: Life Science Alliance Manuscript #LSA-2019-00398-T

Dr. Thomas B Hansen
Aarhus University

Dear Dr. Hansen,

Thank you for submitting your revised manuscript entitled "Non-coding AUG circRNAs constitute an abundant and conserved subclass of circles". Your manuscript was previously reviewed twice at another journal, and the editors shared those reports with us with your permission. You already responded and revised your study in response to the concerns raised by the reviewers upon re-review.

The reviewers who evaluated your study thought that you addressed many of their concerns. They thought that some of your important conclusions are based on negative data and that the mechanistic insight / insight into circRNA biogenesis remains limited. You already outlined that your analyses were controlled with positive and negative controls, providing support for your conclusions even if based on negative data. We appreciate the introduced changes and would be happy to publish your paper in Life Science Alliance pending final revisions, mainly necessary to meet our formatting guidelines:

- I would like to suggest to change the wording in the abstract to "... and they display flanking sequences that suggest an Alu-independent mechanism of biogenesis" to better address reviewer #1's concern.
- Please add a reference to Supplementary Table 3 when discussing HITS-CLIP data (also in the methods section)
- Please add callouts in the manuscript text to Fig7C, Fig S8A, B, C
- Please upload all figure files (also S figures, and without legends) as individual files; please incorporate the S figure legends into the main manuscript docx file
- Please add scale bars to Fig 4F and SFig9/10G
- We are displaying supplementary files in-line in the HTML version of the paper, for which they need to be on a single page only => please split figure S2 and S15 into several figures or upload as a supplementary data set
- Please link your ORCID iD to your profile in our submission system, you should have received a message on how to do so
- Please include a summary blurb, subject categories, author contributions and a COI statement in our submission system

You will be guided to complete the submission of your revised manuscript and to fill in all necessary

information. Please get in touch in case you do not know or remember your login name.

A. FINAL FILES:

B. MANUSCRIPT ORGANIZATION AND FORMATTING:

Sincerely,

April 16, 2019

RE: Life Science Alliance Manuscript #LSA-2019-00398-TR

Dear Dr. Hansen,

Thank you for submitting your Research Article entitled "Non-coding AUG circRNAs constitute an abundant and conserved subclass of circles". It is a pleasure to let you know that your manuscript is now accepted for publication in Life Science Alliance. Congratulations on this interesting work.

DISTRIBUTION OF MATERIALS:

Again, congratulations on a very nice paper. I hope you found the review process to be constructive and are pleased with how the manuscript was handled editorially. We look forward to future exciting submissions from your lab.

Sincerely,
Andrea Leibfried, PhD
Executive Editor
Life Science Alliance